# The Simple One-step stool processing method for detection of Pulmonary tuberculosis: A study protocol to assess the robustness, stool storage conditions and sampling strategy for global implementation and scale-up

Petra de Haas[1]*, Bazezew Yenew[2], Getu Diriba[2], Misikir Amare[2], Andrii Slyzkyi[1], Yohannes Demissie[3], Bihil Sherefdin[3], Ahmed Bedru[3], Endale Mengesha[3], Zewdu Gashu Dememew[3], Abebaw Kebede[2,4,5], Muluwork Getahun[2], Edine Tiemersma[1], Degu Jerene[1]

1 KNCV Tuberculosis Foundation, The Hague, The Netherlands, 2 Ethiopian Public Health Institute, Addis Ababa, Ethiopia, 3 KNCV Tuberculosis Foundation Ethiopia Office, Addis Ababa, Ethiopia, 4 Department of Microbial, Cellular and Molecular Biology, College of Natural and Computational Sciences, Addis Ababa University, Addis Ababa, Ethiopia, 5 Africa CDC, Addis Ababa, Ethiopia

* petra.dehaas@kncvtbc.org

**Data Availability Statement:** No datasets were generated or analysed during the current study. All

## Abstract

### Background

The Xpert MTB/RIF Ultra (Xpert-Ultra) assay provides timely results with good sensitivity and acceptable specificity with stool specimens in children for bacteriological confirmation of tuberculosis (TB). This study aims to optimize the Simple One-Step (SOS) stool processing method for testing stool specimens using the Xpert-Ultra in children and adults in selected health facilities in Addis Ababa, Ethiopia. The study is designed to assess the robustness of the SOS stool method, to help fine-tune the practical aspects of performing the test and to provide insights in stool storage conditions and sampling strategies before the method can be implemented and scaled in routine settings in Ethiopia as well as globally.

### Methods and design

The project "painless optimized diagnosis of TB in Ethiopian children" (PODTEC) will be a cross sectional study where three key experiments will be carried out focusing on 1) sampling strategy to investigate if the Xpert-Ultra *M. tuberculosis* (MTB) -positivity rate depends on stool consistency, and if sensitivity can be increased by taking more than one stool specimen from the same participant, or doing multiple tests from the same stool specimen, 2) storage conditions to determine how long and at what temperature stool can be stored without losing sensitivity, and 3) optimization of sensitivity and robustness of the SOS stool processing method by varying stool processing steps, stool volume, and storage time and conditions of the stool-sample reagent mixture. Stool specimens will be collected from participants (children and adults) who are either sputum or naso-gastric aspiration (NGA) and/

relevant data from this study will be made available upon study completion.

**Funding:** This study protocol is part of the PODTEC (painless optimised diagnosis of TB in ethiopian childern) study and is funded by a private funder. The funder had and will not have a role in study design, data collection and analysis, decision to publish, or preparation of the manuscript.

**Competing interests:** The authors have declared that no competing interests exist.

or stool Xpert-Ultra MTB positive depending on the experiment. Stool specimens from these participants, recruited from 22 sites for an ongoing related study, will be utilized for the POD-TEC experiments. The sample size is estimated to be 50 participants. We will use EpiData for data entry and Stata for data analysis purposes. The main analyses will include computing the loss or gain in the Xpert-Ultra MTB positivity rate and rates of non-determinate Xpert-Ultra test results per experiment compared to the Xpert-Ultra MTB result of stool processed according to the published standard operating procedures for SOS stool processing. The differences in the MTB positivity rate by regarding testing more than one sample per child, and using different storage, and processing conditions, will be also compared to the baseline (on-site) Xpert-Ultra result.

## Introduction

Approximately 1.09 million children globally fall ill with tuberculosis (TB) each year, of whom only 399.000 are notified [1]. Every day, nearly 700 children die from TB, 80% of them before reaching their fifth birthday. Treatment exists that could prevent nearly all these deaths, but less than 5% of children get treatment as childhood TB is difficult to diagnose [2].

Recent WHO guidance on diagnosis and management of TB in children and adolescents recommends stool as non-invasive primary diagnostic specimen for testing with Xpert and Xpert Ultra for a diagnosis of TB among children [3, 4].

This new recommendation was guided by a recent systematic review, which showed that for stool, Xpert Ultra sensitivity against microbiological reference standard (MRS) was 53% in children aged 0 to 9 years, the sensitivity being higher in children younger than 1 year (65%), and lower in those aged 1 to 4 years (43%). Specificity was 96% to 98% [5]. Together with the Global laboratory initiative (GLI), WHO recently provided recommendations for the stool processing methods to be used, as there is a large variety on protocols for stool processing, with differences in reagents and methods of homogenization, filtering, and other steps leading to more complexity and a high heterogeneity in sensitivity [6]. One of the two stool processing methods recommended by WHO/GLI is the Simple One-Step (SOS) stool processing method, which is developed by KNCV Tuberculosis Foundation (KNCV) in collaboration with Ethiopian Public Health Institute (EPHI) [6, 7]. The SOS stool method uses similar steps as sputum Xpert and Xpert-Ultra testing and does not require additional materials or equipment other than an applicator to pick the correct stool amount for testing [6, 7]. Since the method is as simple as sputum testing, it can be performed at any site where a GeneXpert instrument is functional after providing minimal training to the staff involved in Xpert or Xpert-Ultra testing [6, 7].

To gain more knowledge and in-depth experience on how the SOS stool processing method with Xpert-Ultra would behave if included in the routine diagnostic pathway for (childhood) TB and rolled out under the national TB program, we aim to further test and optimize the SOS processing method for the detection of TB in stool by Xpert-Ultra and its ability to tolerate perturbations (robustness). The study will also help to fine-tune standard operating procedures (SOPs) for the SOS stool method.

## Materials and methods

### Study setup and period

This will be a cross-sectional study that will consist of a series of experiments on consecutive stool specimens collected from children and adults that are either sputum/NGA and/or stool

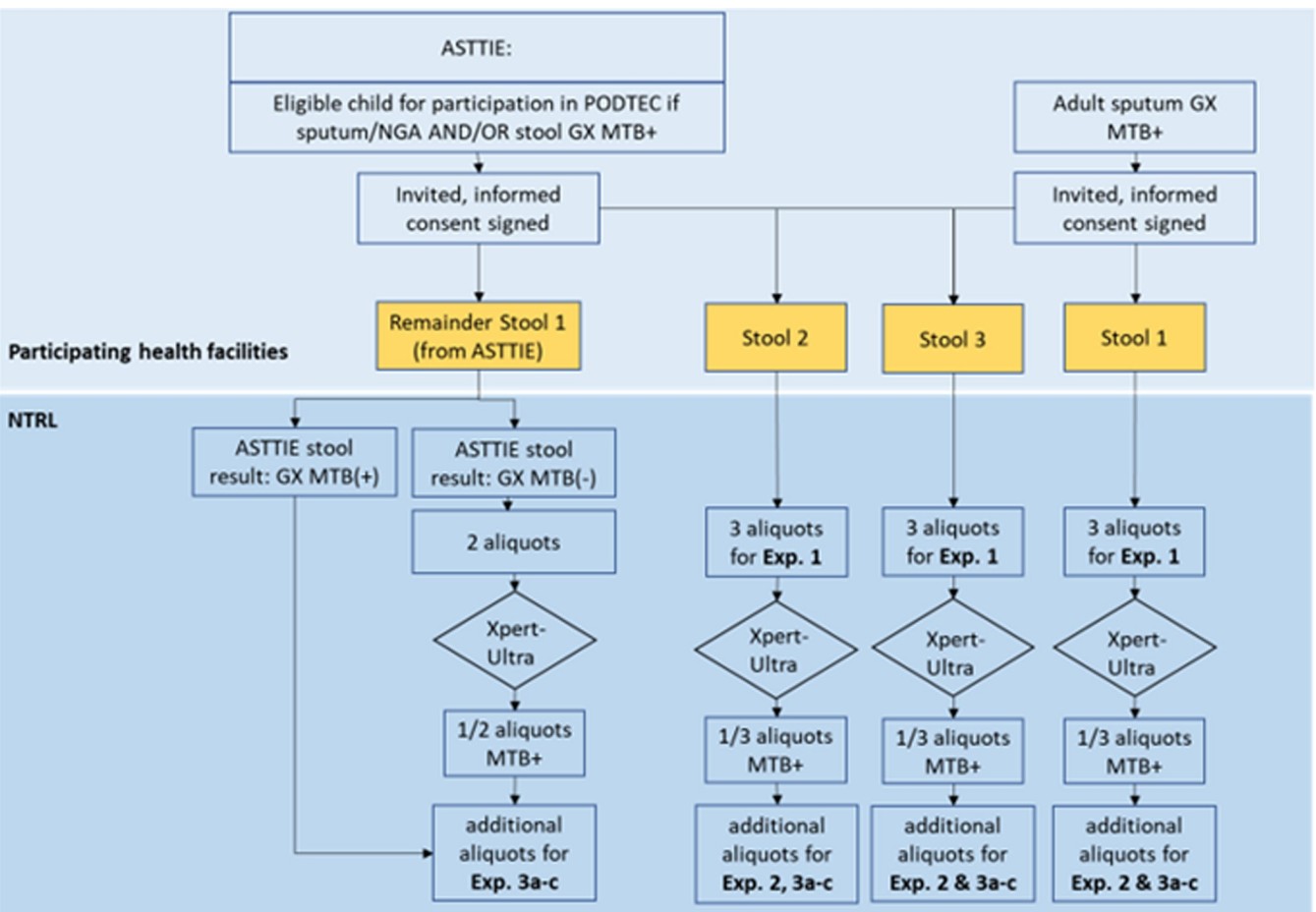

**Fig 1. Study flow diagram representing steps from inclusion of adults and children to arrival of the stool samples at EPHI and their allocation to the different experiments.** * Site of action is depicted with blue background boxes; and stool samples are in yellow boxes. Abbreviations used: ASTTIE; Alternatives to Sputum Testing for Tuberculosis in Indonesia and Ethiopia, PODTEC: Painless Optimized Diagnosis of TB in Ethiopian Children, NTRL: National TB Reference Laboratory, NGA: Naso-Gastric Aspiration, MTB: *M.tuberculosis*, GX; GeneXpert, Exp.: Experiment.

Xpert-Ultra MTB positive. The study will be conducted in multiple health care facilities (>20) in Addis Ababa, Ethiopia. The facilities have a relatively high number of TB patients and have experience with participation in research. Children are being recruited for another related study, that assesses the diagnostic accuracy of Xpert stool testing using the SOS stool processing method, called Alternatives to Sputum Testing for Tuberculosis in Indonesia and Ethiopia (ASTTIE), see Fig 1. Therefore, MTB positive children who will be identified in the ASTTIE study will also be used in the current study (PODTEC study). We will also recruit adults with MTB detected in sputum from the same facilities. The study was originally planned to be carried out till the end of 2020. However, due to the COVID-19 pandemic, the study period has been extended.

## Study population

Children aged ≤10 years from the ASTTIE study who are sputum/NGA and/or stool Xpert-Ultra MTB positive, and consecutive sputum Xpert MTB-positive adults presenting in the selected health facilities will constitute the study population. Eligible persons, or their caregivers will be requested to sign (parental) consent or assent, depending on the age of the

participant. The exclusion criteria include being critically sick i.e., those who are in coma, terminally ill due to chronic debilitating co-morbidities, or other conditions determined to be "critical" by the treating physician, being on TB treatment for longer than 5 days at the time of recruitment, and refusal to sign the informed consent.

## Participant enrolment and stool collection

For children, the facility coordinator from the ASTTIE project will daily retrieve stool Xpert-Ultra results from the study sites and checks for eligibility. Parents of eligible children and eligible adults will be asked for informed consent for participation in this study, see S1 Appendix. For children, this is an additional consent to the consent already provided for the ASTTIE study. After enrolment in the study, for children, the remainder of the initial stool specimen (stool 1) will be collected and transported to EPHI. Participants will be provided with two (children) or three (adults) large stool containers to allow collection of at least 30 grams of stool on consecutive days. They will be instructed on how to collect and store the specimens till delivery at the site. Three appointments will be made to submit the additional stool specimens. When the specimens are submitted, information will be collected on the stool submission form S2 Appendix about the date and time of collection at the household, storage conditions at the household and during transport and date and time of arrival at the site. To maximize the likelihood of finding MTB in the stool, the additional stool specimens should be collected within 5 days after the participant's TB treatment starts. Participants will be reimbursed for travel costs.

The stool specimens will be kept in a cold chain at the site until they are transported to EPHI on the same day. The site will inform the study coordinator which will assign a dedicated transporter for this research purpose. The time between contacting the study coordinator and the pick-up of the specimens is expected to be within 2 hours after delivery on-site, which means that the specimens should arrive on the same day of collection at EPHI. After arrival at EPHI, the dedicated laboratory technician will be ready to receive and register the specimens and start the cascade of experiments as shown in the specimen flow diagram Fig 1.

## Design of the experiments

In total three experiments are designed and the lay out is depicted in Figs 2 and 3.

**Experiment 1: Stool sampling strategy.** This experiment will investigate if, and by how much, the positivity rate of Xpert-Ultra on stool increases when more than one specimen from the same participant is tested. It also indicates how homogeneous the Mycobacteria are distributed within the stool specimens and across different stools. Furthermore, it will provide insight in repeat testing if required due to unsuccessful test result, whether to advise to perform the repeat test from the same stool or from a fresh stool specimen. This will be done by testing three stools from the same participant collected during consecutive days (Experiment 1a) as well as three aliquots (North, South and East/West) from the same stool specimen (Experiment 1b). For children, two aliquots will be collected from the first stool specimen Fig 1. Aliquots will be tested using Xpert-Ultra, totaling a maximum of nine Xpert-Ultra tests per participant.

**Experiment 2: Stool sample storage conditions.** This experiment will investigate how long and under which conditions stool can be kept without losing sensitivity to detect TB on Xpert-Ultra or increasing rates of unsuccessful tests. This is done by testing multiple aliquots taken from a known Xpert-Ultra MTB-positive stool specimen after storing aliquots from that Xpert-Ultra MTB-positive stool at three temperatures; a) refrigerator 2–8˚C, b) room temperature 20–22˚C, and c) incubator 37˚C and at four time periods; 2, 3, 5 and 10 days. These experiments will be done using aliquots remaining from the second and third stool specimens

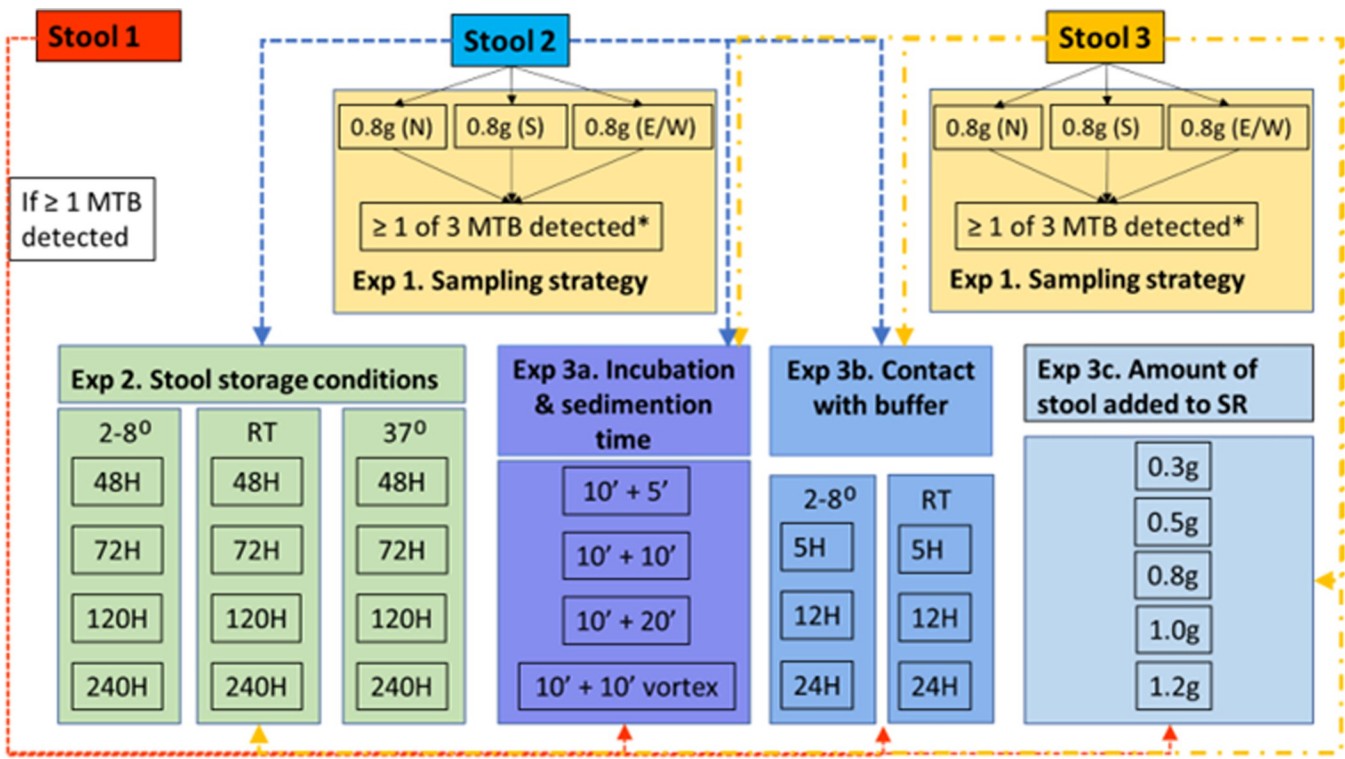

**Fig 2. Overview of experiments.** Arrows indicate what experiments are done with the different stool specimens. Note that for adult participants, in principle all experiments will be done on all three stool samples, provided that enough stool is collected per bowel movement. Abbreviations used: MTB, M. tuberculosis; N, North; S, South; E/W, East or West; Exp, experiment; RT, room temperature; H, hours; SR, sample reagent; g, gram.

collected for Experiment 1. These stool specimens are expected to be larger in size and probably have the shortest time interval between collection at the site and preparation at EPHI.

**Experiment 3: Optimization and evaluating robustness of the SOS stool processing method.** This experiment consists of a series of sub-experiments that will investigate if the SOS stool processing method can be further optimized to increase its recovery rate to detect TB. Although the SOS stool processing method is simple and contains minimal processing steps, certain steps might still be adapted to see if this influences the test's sensitivity. This is done by testing multiple aliquots taken from a known Xpert-Ultra MTB-positive stool specimen processed using slightly different approaches. The first sub-experiment (3a) varies the incubation time and shaking method during the processing of stool. The second sub-experiment (3b) assesses the optimum and maximum time and temperature for keeping the processed stool-sample reagent mixture before Xpert-Ultra testing is conducted on the different incubation steps. The third sub-experiment (3c) assesses the optimum and maximum stool volume.

Stool consistency and bacterial load are two important factors that might influence the outcome of the experiments. Therefore, specimens with different consistency and bacterial load will be included in all experiments.

If the Xpert-Ultra result is unsuccessful (i.e., the result is "invalid" or "error"), the test will NOT be repeated as this is part of the study outcome.

The SOS stool method's comprehensive instructions can be found on KNCV website [8]. In S3 Appendix a schematic overview of the SOS stool method is provided. Depending on the stool consistency the protocol for solid stool or liquid stool is followed.

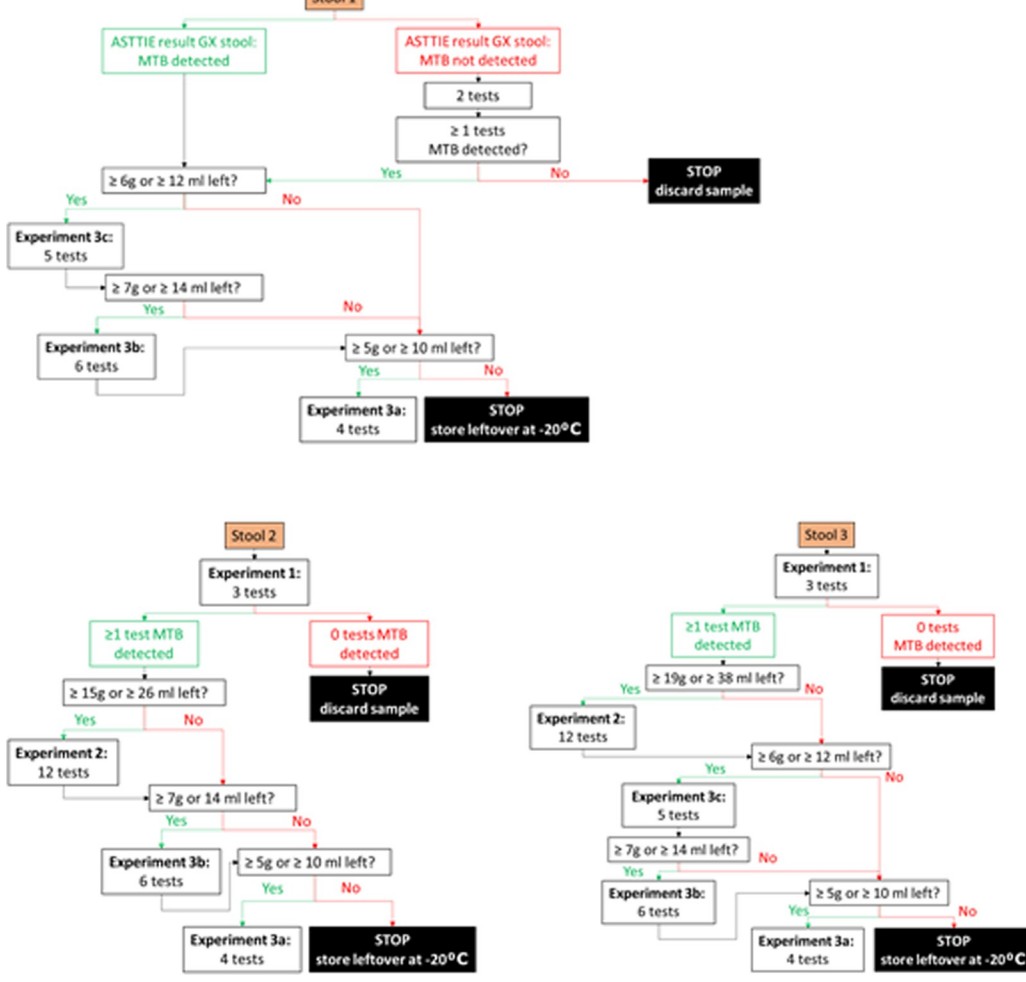

**Fig 3. Overview of assigning stool to the experiments outlined in Fig 2.** Note that for adults, in principle, all experiments will be conducted if there is enough stool available per sample. Abbreviations used: ASTTIE; Alternatives to Sputum Testing for Tuberculosis in Indonesia and Ethiopia, MTB: *M. tuberculosis*, GX; GeneXpert, Exp.: Experiment; g, gram: ml, milliliter. Explanation of experiment numbers: Experiment 1: Stool sampling strategy; Experiment 2: Stool sample storage conditions; Experiment 3a: incubation time and shaking method; Experiment 3b: time and temperature for keeping the processed stool-sample reagent mixture; Experiment 3c: stool volume.

## Variables and outcome measures

The primary outcome measures will be the rate of Xpert-Ultra indeterminate test results. The secondary outcome measures will be the Xpert-Ultra MTB (semi-)quantitative result and positivity rate of stool specimens processed using the SOS stool processing method. Values of these outcomes obtained under the different experimental conditions will be compared to the baseline (on-site) Xpert-Ultra MTB positive test result of stool processed using the SOS stool processing method and to the values obtained under per current protocol experimental conditions [6, 7].

## Sample size

At the selected health facilities, a maximum of 750 children with presumptive TB will be enrolled in ASTTIE during the recruitment period of this study. With Xpert-Ultra MTB-

| Shifting from + to - | Shifting from - to + | | | | | | | | | | |
|---|---|---|---|---|---|---|---|---|---|---|---|
| | 0% | 1% | 2% | 3% | 4% | 5% | 10% | 15% | 20% | 25% | 30% |
| 0% | Infinite | 783 | 391 | 260 | 194 | 155 | 77 | 50 | 37 | 29 | 24 |
| 1% | 783 | Infinite | 2353 | 783 | 434 | 292 | 105 | 62 | 44 | 33 | 27 |
| 2% | 391 | 2353 | Infinite | 3923 | 1175 | 609 | 145 | 77 | 51 | 38 | 30 |
| 3% | 260 | 783 | 3923 | Infinite | 5492 | 1568 | 206 | 96 | 61 | 43 | 34 |
| 4% | 194 | 434 | 1175 | 5492 | Infinite | 7062 | 303 | 121 | 72 | 50 | 38 |
| 5% | 155 | 292 | 609 | 1568 | 7062 | Infinite | 469 | 155 | 85 | 57 | 42 |
| 10% | 77 | 105 | 145 | 206 | 303 | 469 | Infinite | 783 | 234 | 120 | 77 |
| 15% | 50 | 62 | 77 | 96 | 121 | 155 | 783 | Infinite | 1097 | 312 | 155 |
| 20% | 37 | 44 | 51 | 61 | 72 | 85 | 234 | 1097 | Infinite | 1411 | 391 |
| 25% | 29 | 33 | 38 | 43 | 50 | 57 | 120 | 312 | 1411 | Infinite | 1725 |
| 30% | 24 | 27 | 30 | 34 | 38 | 42 | 77 | 155 | 391 | 1725 | Infinite |

: number can be reached if all 3 stool samples are used in the experiment

: number can be reached if 2 stool samples are used in the experiment

: number can be reached if 1 stool sample is used in the experiment

**Fig 4. Required sample size as estimated using McNemar's two-sample paired proportions test using Stata SE v.** 17.0, assuming discordance in pairs of stool aliquots.

positivity rate of 5%, up to 32 children will be available for the optimization exercises. We aim to supplement this with up to 50 participants by also recruiting adults from the health facilities participating in ASTTIE, as described above, as the SOS stool test with Xpert has shown to be also useful for adults who cannot spontaneously expectorate sputum [9]. Thus, for the experiments, we will have stool specimens for around 50 individuals available, totaling a maximum of 150 stool specimens.

Fig 4 indicates the minimum rates of conversion (Xpert MTB- to Xpert MTB+), respectively reversion (Xpert MTB+ to Xpert MTB-), that can be detected with statistical significance with this sample size.

## Analysis plan and data collection

Data collected will include age and sex of the participant, TB suggestive symptoms and TB contact history, Xpert-Ultra result for the initial stool (children only) and sputum/NGA specimen, date, and place (participant's home or health facility) of stool collection, date of stool receipt at the NTRL, stool storage and transport temperature until receipt at the NTRL, and stool consistency. Detailed information on the experiments' conditions will also be collected as well as the cycle threshold (Ct) values for all probes and error codes in case of errors.

## Data entry, storage, and management

Each stool specimen will be submitted to EPHI together with a stool submission form S2 Appendix. Details when conducting the experiment are collected on the experiment form S4 Appendix. The forms are stored at EPHI. All data will be entered into pre-structured EpiData

files (EpiData version 3.1; www.epidata.dk). A random 10% of the data will be re-entered in a separate file to check the quality of data entry. If more than 3% of errors are found in key variables (experiment conditions and Xpert result), full double data entry will be conducted.

## Data interpretation

The main study outcome is the semi-quantitative Xpert Ultra result as provided by the GeneXpert instrument (trace, very low, low, medium, high, error, invalid or no result), interpreted as per the manufacturer's instructions. The individual probes' Ct values are the main quantitative study outcomes. We consider higher Ct values to represent lower bacillary loads, as these indicate that more PCR cycles are needed to reach the threshold of MTB detected. Error codes will be interpreted following the manufacturer's guidance to understand the likely cause of the error.

## Statistical analysis

Statistical analysis will be performed by the STATA/SE (version 15; StataCorp) statistical software package. The Xpert-Ultra stool result from each aliquot will be compared with the baseline Xpert-Ultra. Trends in the proportion of specimens being MTB-positive and the proportion of specimens with unsuccessful results over e.g., increasing storage time or temperature, or increasing amounts of stool added, will be analyzed using Wilcoxon-like test for trend of across ordered groups using nptrend [10]. Logistic regression will be applied to assess factors associated with stool MTB positivity and unsuccessful test outcomes. We will assess if there is indication that stool specimens are nested into individual participants, just as aliquots of one stool specimen may be nested into that stool specimen, by comparing the outcomes of simple (multivariate) logistic regression to the outcomes of multilevel mixed-effects logistic regression using the likelihood-ratio test to determine the best model fit.

## Ethical considerations

The study obtained ethical clearance from the Review Boards of the Ethiopian Public Health Institute (EPHI-IRB) (Protocol no EPHI-IRB-234-2020). The project will follow the routine procedure of patients' recruitment into studies, follow-up, and analysis as well as drawing of conclusions. Informed parental consent will be obtained from the children's legal guardians. Participants' information will be kept confidential, and the digital files used for analysis will only have the PODTEC laboratory code and ASTTIE unique person identification code (UPIC) and will not contain any names or other personal identifying information of the participant. Participants' information will be kept confidential, and the digital files used for analysis will only have the PODTEC laboratory code and the ASTTIE UPIC and will not contain any names or other personal identifying information of the participant. The study results will be shared with the national TB program and stakeholders to the benefit of further roll out of the test in a routine Ethiopian setting. The results will also be disseminated in peer-reviewed scientific journals.

## Discussion

This is the first study protocol to be published in which the sampling strategy and robustness of a stool processing method will be investigated. Based on the experiment's findings, certain steps in the current SOP of the SOS stool method might be adjusted. The experiments will be performed using specimens from bacteriologically confirmed TB patients, so for the patients for whom the test will be used in practice in a country with a relatively high TB burden. Stool

specimens will not be spiked with MTB-complex bacteria as in some other studies [11, 12]. We expect that the MTB distribution is different in the stool specimens from TB patients than in the spiked stools. Moreover, we will include children who have mostly have paucibacillary TB and who will benefit most from stool Xpert or Xpert-Ultra testing, as they cannot easily provide sputum. The study population will be drawn from locations where the test is expected to be conducted in the future, providing more realistic insights in the possibility of implementing the method in routine and collection of multiple specimens.

The experiments are based on the controlled simulation of plausible scenarios, i.e. situations that can occur in practice, such as long transit times at high temperatures, long contact time of stool with the sample reagent before Xpert-Ultra testing, variation in the stool portion size picked for processing, and variation in stool processing. The outcome will provide insights of the robustness of the method and will show how far certain steps can be stretched. It will provide practical outcomes that will enable the laboratory personnel and healthcare professionals involved in the stool testing to implement the most optimal protocol.

The main results will be presented both at local and international scientific meetings. The results will also be disseminated in the form of peer reviewed publications and as policy briefs. Key audiences for the dissemination will include global scientific advisory group members, local technical advisory committee (TAC) members and NTP. Study host communities will also be informed about the key results of the study through appropriate popular media.

## Supporting information

**S1 Appendix.**
(PDF)

**S2 Appendix.**
(PDF)

**S3 Appendix.**
(TIF)

**S4 Appendix.**
(PDF)

## Acknowledgments

We would like to acknowledge the Ethiopian Public Health Institute and KNCV Tuberculosis Foundation. We would also like to thank all healthcare facilities that will participate in the study by recruiting the study participants. We would further like to acknowledge Mamush Sahile from KNCV Ethiopia for his assistance in the study.

## Author Contributions

**Conceptualization:** Petra de Haas, Bazezew Yenew, Andrii Slyzkyi, Edine Tiemersma, Degu Jerene.

**Data curation:** Bazezew Yenew, Getu Diriba, Misikir Amare, Bihil Sherefdin.

**Formal analysis:** Petra de Haas, Bazezew Yenew, Edine Tiemersma.

**Funding acquisition:** Petra de Haas, Ahmed Bedru, Edine Tiemersma, Degu Jerene.

**Investigation:** Petra de Haas, Bazezew Yenew, Getu Diriba, Andrii Slyzkyi, Bihil Sherefdin, Endale Mengesha, Edine Tiemersma, Degu Jerene.

**Methodology:** Petra de Haas, Bazezew Yenew, Getu Diriba, Misikir Amare, Andrii Slyzkyi, Bihil Sherefdin, Endale Mengesha, Zewdu Gashu Dememew, Abebaw Kebede, Muluwork Getahun, Edine Tiemersma, Degu Jerene.

**Project administration:** Bazezew Yenew, Yohannes Demissie, Bihil Sherefdin, Ahmed Bedru, Endale Mengesha, Zewdu Gashu Dememew, Degu Jerene.

**Resources:** Ahmed Bedru, Degu Jerene.

**Software:** Edine Tiemersma.

**Supervision:** Petra de Haas, Bazezew Yenew, Getu Diriba, Misikir Amare, Andrii Slyzkyi, Yohannes Demissie, Bihil Sherefdin, Endale Mengesha, Zewdu Gashu Dememew, Abebaw Kebede, Muluwork Getahun, Edine Tiemersma, Degu Jerene.

**Validation:** Petra de Haas, Edine Tiemersma.

**Visualization:** Petra de Haas, Edine Tiemersma.

**Writing – original draft:** Petra de Haas, Bazezew Yenew, Edine Tiemersma, Degu Jerene.

**Writing – review & editing:** Petra de Haas, Bazezew Yenew, Getu Diriba, Misikir Amare, Andrii Slyzkyi, Yohannes Demissie, Bihil Sherefdin, Ahmed Bedru, Endale Mengesha, Zewdu Gashu Dememew, Abebaw Kebede, Muluwork Getahun, Edine Tiemersma, Degu Jerene.

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
