## [Decision Letter · Decision Letter 0]

23 May 2022

PONE-D-22-03274The Simple One-step stool processing method for detection of Pulmonary tuberculosis: a study protocol to assess the robustness, stool storage conditions and sampling strategy for global implementation and scale-upPLOS ONE

Dear Dr. de haas,

Thank you for submitting your manuscript to PLOS ONE. After careful consideration, we feel that it has merit but does not fully meet PLOS ONE’s publication criteria as it currently stands. Therefore, we invite you to submit a revised version of the manuscript that addresses the points raised during the review process. The study protocol is presented well and well throughout. But it does need improvement and clarity. As suggested by the reviewer the flow charts could be simple. I hope the comments would help you improve the manuscript and I look forward for your resubmitted manuscript.

We look forward to receiving your revised manuscript.

Kind regards,

Padmapriya P Banada, PhD

Academic Editor

PLOS ONE

Journal Requirements:

2. Please include a separate caption for each figure in your manuscript.

3. Please ensure that you refer to Figure 1, 2, 3 and 4 in your text as, if accepted, production will need this reference to link the reader to the figure.

4. Please upload a copy of Supporting Information Figure S1, S2, S3 and S4 which you refer to in your text on page 6, 7 and 8.

Additional Editor Comments:

Dear authors,

The study protocol is presented well and well thoughtout. But it does need improvement and clarity. As suggested by the reviewer the flow charts could be simple. I hope the comments would help you improve the manuscript and I look forward for your resubmitted manuscript.

Best

Priya

Reviewers' comments:

Reviewer's Responses to Questions

**Comments to the Author**

1. Does the manuscript provide a valid rationale for the proposed study, with clearly identified and justified research questions?

Reviewer #1: Yes

Reviewer #2: Yes

2. Is the protocol technically sound and planned in a manner that will lead to a meaningful outcome and allow testing the stated hypotheses?

Reviewer #1: Partly

Reviewer #2: Yes

3. Is the methodology feasible and described in sufficient detail to allow the work to be replicable?

Reviewer #1: No

Reviewer #2: Yes

4. Have the authors described where all data underlying the findings will be made available when the study is complete?

Reviewer #1: Yes

Reviewer #2: Yes

5. Is the manuscript presented in an intelligible fashion and written in standard English?

Reviewer #1: No

Reviewer #2: Yes

6. Review Comments to the Author

You may also provide optional suggestions and comments to authors that they might find helpful in planning their study.

Reviewer #1: The protocol proposed by De Haas and colleagues aims to assess different components of stool collection and processing which could influence detection of MTB by Xpert Ultra. The flow charts are rather complex but have clearly been thought through carefully.

My main points of criticism are the following:

- The background information needs to be more balanced to accurately summarize current evidence of different stool processing methods for TB diagnosis (see details below)

- I am uncertain about the sample size calculation and whether the estimated sample size will be able to detect a meaningful difference between the experiments and the baseline method.

Abstract

General: The abstract uses the terms “samples” and “sampling” loosely to mean specimen, statistical sampling and testing different parts of a sample. I recommend reviewing the entire abstract to use more accurate terminology.

Other:

Line 21: “Xpert MTB/RIF Ultra (Xpert-Ultra) provides timely results with good sensitivity and acceptable specificity with stool samples in children for bacteriological confirmation of tuberculosis (TB).” There are actually no data on Ultra on stool in children- only Xpert MTB/RIF.

Line 41: Reword: “The sample size is estimated to be 50 participants.”

Line 43-47: Reword: “We will use EpiData for data entry and Stata for data analysis purposes. The main analyses will include computing the loss or gain in the Xpert-Ultra MTB positivity rate compared to ..., and rates of unsuccessful test results. The differences in the positivity rate by regarding testing more than one sample per child, and using different storage, and processing conditions, will be compared to the baseline (on-site) Xpert-Ultra result.”

Introduction

WHO has made no recommendation on stool as a sample for use with Ultra due to lack of evidence. The recommendation for stool is for Xpert MTB RIF only, and it is based on low certainty of evidence.

Lines 71-72: “This shows that there is a lack of standardized stool preparation and testing protocols and warrants the optimization and standardization of the stool processing methods that can be used at the decentralized level.”

Suggest rewording: “The lack of standardized stool preparation and testing protocols warrants the optimization and standardization of stool processing methods that can be used at the decentralized level

Lines 81-82: I suggest rewording: A pilot study conducted in multiple laboratories across Ethiopia demonstrated acceptably low rates of unsuccessful test results (6%).

Lines 83-84: “Furthermore, a head-to-head comparison study, in which the SOS method is compared to other stool processing methods showed similar sensitivity and specificity.” I think it is rash to make this statement based on a laboratory spiking study which tested a small number of samples. Even so, the Walters-centrifugation method appeared superior in detecting BCG at lower concentrations, which is relevant in the case of young children/infants, who have more extreme forms of paucibacillary disease and for whom stool is more attractive than for older children and adolescents. I would favour a more balanced summary of the quoted study.

Methods

Enrolment: I would strongly advise enriching for young children and infants <2 years of age, as this is the group for whom stool-based diagnosis is most relevant. This is also the group more likely to have very low bacillary concentrations in sputum and hence stool. A stool-based method that can detect TB in older children with adult-type cavitary TB and other forms of TB with higher bacillary loads (many of these children will be able to produce sputum) is not as relevant.

Lines 192-196 : The sentence is grammatically incorrect. Please clarify if the primary outcome is the Ultra positivity rate of the index experiment vs the on-site Ultra stool result? Is the secondary outcomes measure also a comparison of the index experiment vs on-site Ultra stool? Please edit accordingly.

I am not an expert in statistics, but I have some concerns regarding the sample size calculation and the effect size that such a sample size will be able to achieve. In the footnote to the figure, the authors say that the SS calculations do not take into account for correlation between stool samples from the same individual. Surely, this should be considered? Secondly, even assuming that all the samples are collected, the minimum difference in detection (from negative to positive) that will be measured with statistical significance is 10%. Does that mean that ANY of the experiments will be able to detect 10% more TB than the baseline test? Is that meaningful? Should the sample size not be calculated to achieve an meaningful increase in detection for every experiment? As I am not an expert in this, I think the statistical methods should be reviewed by a statistician, and clarified for a non-expert readership.

Discussion

Lines 262-263. I know that other stool processing methods have undergone similar pre-clinical testing, but the protocols were not published. So rather say that this is the first protocol to be published...

Line 265: The experiments will actually be conducted on confirmed TB cases based on the inclusion criteria, not presumptive TB cases.

Line 266-267: There are a number of published studies assessing stool-based TB diagnosis which have used clinical samples (not only spiked samples).

Minor:

Line 70: rather quite- redundancy

Line 212: Suggest rewording: “Data collected will include age...”

Line 214: “of stool of collection”- delete second “of”

Reviewer #2: This study protocol is very elaborative and well designed. The authors here would like to emphasize on the need of a simple and standard protocol for Mtb diagnosis from stool samples. Their study protocol is easy to understand with enough supporting information. I would like to provide my concerns/suggestions listed below.

1. Which sample will be used as baseline? the stool or the sputum or both?

2. Will the samples be collected on two/three consecutive days from the study population or there will be some duration between two samples collected from the same person? My suggestion will be to include this information in the study protocol as well as in the consent form.

3. Are you going to enroll Mtb negative population for this study as a control group?

4. For experiment 2, to study the effect of storage conditions, you have not proposed to freeze any samples. I believe the addition of freezing as a stool storage temperature and then study the effect of freezing on Mtb detection using your proposed SOS method would add value to your studies and also help others in future.

7. PLOS authors have the option to publish the peer review history of their article (what does this mean?). If published, this will include your full peer review and any attached files.

Reviewer #1: No

Reviewer #2: No

---

## [Author Response · Author response to Decision Letter 0]

4 Jul 2022

Feedback to the reviewers

Reviewer #1: The protocol proposed by De Haas and colleagues aims to assess different components of stool collection and processing which could influence detection of MTB by Xpert-Ultra. The flow charts are rather complex but have clearly been thought through carefully.

My main points of criticism are the following:

- The background information needs to be more balanced to accurately summarize current evidence of different stool processing methods for TB diagnosis (see details below)

- I am uncertain about the sample size calculation and whether the estimated sample size will be able to detect a meaningful difference between the experiments and the baseline method. 

We thank the reviewer for these considerations and refer to our answers below, in which we have attempted to address these concerns as much as follows.

Abstract

General: The abstract uses the terms “samples” and “sampling” loosely to mean specimen, statistical sampling and testing different parts of a sample. I recommend reviewing the entire abstract to use more accurate terminology.

We agree with the reviewer that the use of the word “sample” for biological specimens and for statistical sample size might lead to confusion among readers. Therefore, we consistently changed the word “sample” to “specimen” when it was referring to a biological specimen (either stool or NGA/sputum) throughout the manuscript. We believe that we already were consistent in using the word “aliquot” for testing different parts of the stool specimen. We left the word “sample” in for the following: 1) when referring to the sample reagent as Cepheid has given this name to their buffer; and 2) when referring to sample size, as these two words were always used in combination. We kept in the term “sampling strategy” as these two words were always used in combination.

Other:

Line 21: “Xpert MTB/RIF Ultra (Xpert-Ultra) provides timely results with good sensitivity and acceptable specificity with stool samples in children for bacteriological confirmation of tuberculosis (TB).” There are actually no data on Ultra on stool in children- only Xpert MTB/RIF. 

The reviewer is correct that at the time the protocol was drafted there was no published data on the use of Xpert-Ultra on stool. However, there was un-published data availble from ongoing studies conducted by KNCV and by other reasearchers that provided promising results. This made us decide to use Xpert-Ultra within this study, as this will also be expected to be used in future routine practise. Meanwhile, data has been published and recommendations have been provided by WHO for the use of Xpert-Ultra. We changed the introduction accordingly. Note that we added the recently published updated guideline on the management of tuberculosis in children and adolescents as a reference and the GLI practical manual on stool processing methods in which the SOS stool method is one of the two recommended methods.

Line 41: Reword: “The sample size is estimated to be 50 participants.”

Indeed, this sentence was phrased incorrectly. We changed the sentence as suggested to “The sample size is estimated to be 50 participants.”

Line 43-47: Reword: “We will use EpiData for data entry and Stata for data analysis purposes. The main analyses will include computing the loss or gain in the Xpert-Ultra MTB positivity rate compared to ..., and rates of unsuccessful test results. The differences in the positivity rate by regarding testing more than one sample per child, and using different storage, and processing conditions, will be compared to the baseline (on-site) Xpert-Ultra result.”

We thank the reviewer for this suggestion, it clarifies better what our standard is. We changed the text as suggested: The main analyses will include computing the loss or gain in the Xpert-Ultra MTB positivity rate and rates of non-determinate Xpert-Ultra test results per experiment compared to the Xpert-Ultra MTB result of stool processed according to the published standard operating procedures for SOS stool processing. The differences in the MTB positivity rate by regarding testing more than one sample per child, and using different storage, and processing conditions, will be also compared to the baseline (on-site) Xpert-Ultra result”.

Introduction

WHO has made no recommendation on stool as a sample for use with Ultra due to lack of evidence. The recommendation for stool is for Xpert MTB RIF only, and it is based on low certainty of evidence. 

See our reply to the comment above in which we explained that evidence on the use of Xpert-Ultra on stool recently became available through the published updated guideline on management of tuberculosis in children and adolescents, we revised the text accordingly.

Lines 71-72: “This shows that there is a lack of standardized stool preparation and testing protocols and warrants the optimization and standardization of the stool processing methods that can be used at the decentralized level.”

Suggest rewording: “The lack of standardized stool preparation and testing protocols warrants the optimization and standardization of stool processing methods that can be used at the decentralized level

We agree with suggestion of the reviewer and changed the text to: “The lack of standardized stool preparation and testing protocols warrants the optimization and standardization of stool processing methods that can be used at the decentralized level” as suggested.

Lines 81-82: I suggest rewording: A pilot study conducted in multiple laboratories across Ethiopia demonstrated acceptably low rates of unsuccessful test results (6%). 

Thank you for this suggestion. As we decided to update the introduction section with new presented in the WHO guidelines (see above), we decided to remove this sentence from the introduction section.

Lines 83-84: “Furthermore, a head-to-head comparison study, in which the SOS method is compared to other stool processing methods showed similar sensitivity and specificity.” I think it is rash to make this statement based on a laboratory spiking study which tested a small number of samples. Even so, the Walters-centrifugation method appeared superior in detecting BCG at lower concentrations, which is relevant in the case of young children/infants, who have more extreme forms of paucibacillary disease and for whom stool is more attractive than for older children and adolescents. I would favour a more balanced summary of the quoted study. 

Thank you for pointing out that we referred to the wrong paper. Jasumback’s paper was indeed using spiked samples and did not describe a head-to-head comparison study. Instead, we had meant to refer to the head-to-head comparison study done by FIND and TB-Speed, which is included in the most recent Guideline. We have corrected the reference.

Methods

Enrolment: I would strongly advise enriching for young children and infants <2 years of age, as this is the group for whom stool-based diagnosis is most relevant. This is also the group more likely to have very low bacillary concentrations in sputum and hence stool. A stool-based method that can detect TB in older children with adult-type cavitary TB and other forms of TB with higher bacillary loads (many of these children will be able to produce sputum) is not as relevant.

We agree with the reviewer that TB diagnosis based on stool instead of sputum is most relevant for young children as these cannot produce sputum. Our experience is that children up to 10 years old experience difficulty with sputum production. There are, however, also adult groups, such as PLHIV, that cannot easily produce a spontaneous sputum sample and for these groups, stool can be an alternative sample, as we recently showed in a pilot implementation project in Vietnam. We have now included this reference in lines 201. 

Lines 192-196: The sentence is grammatically incorrect. Please clarify if the primary outcome is the Ultra positivity rate of the index experiment vs the on-site Ultra stool result? Is the secondary outcomes measure also a comparison of the index experiment vs on-site Ultra stool? Please edit accordingly.

We thank the reviewer for remarking this. We have changed the text as follows: “The primary outcome measures will be the rate of Xpert-Ultra indeterminate test results”. The secondary outcome measure will be the Xpert-Ultra MTB (semi-)quantitative result and positivity rate of stool specimens processed using the SOS stool processing method. Values of these outcomes obtained under the different experimental conditions will be compared to the baseline (on-site) Xpert-Ultra MTB positive test result of stool processed using the SOS stool processing method and to the values obtained under per current protocol experimental conditions (6, 7).

I am not an expert in statistics, but I have some concerns regarding the sample size calculation and the effect size that such a sample size will be able to achieve. In the footnote to the figure, the authors say that the SS calculations do not take into account for correlation between stool samples from the same individual. Surely, this should be considered? Secondly, even assuming that all the samples are collected, the minimum difference in detection (from negative to positive) that will be measured with statistical significance is 10%. Does that mean that ANY of the experiments will be able to detect 10% more TB than the baseline test? Is that meaningful? Should the sample size not be calculated to achieve an meaningful increase in detection for every experiment? As I am not an expert in this, I think the statistical methods should be reviewed by a statistician, and clarified for a non-expert readership.

Several of the authors are senior epidemiologists and thus have some statistical education, but we are consulting a statistician for this now. We have used the Statulator (http://statulator.com/SampleSize/ss2PP.html#) which calculates the sample size needed for paired proportions, and then calculated how many observations are needed if x% of the population shifts from MTB+ to MTB-, or, alternatively from a meaningful result (MTB+/MTB-) to an indeterminate result (error, invalid, no result). Thus, this considers the fact that one measurement from the same person (or stool) is paired to another measurement from that same person. Hence, the footnote under the figure is not correct and we removed the footnote. 

We do however agree that the sample size may be too small for the sampling experiment as the positivity rate may not drop/increase with 20% from baseline experiment (or per-protocol experiment). It may probably be sufficient for the robustness experiments, as (initial) indeterminate rates tend to be quite high when deviating from the protocol.

Discussion

Lines 262-263. I know that other stool processing methods have undergone similar pre-clinical testing, but the protocols were not published. So rather say that this is the first protocol to be published...

We are pleased the reviewer is pointing this out, we added “to be published” to the sentence as suggested.

Line 265: The experiments will actually be conducted on confirmed TB cases based on the inclusion criteria, not presumptive TB cases.

This is indeed incorrect, and we thank the reviewer for this correction. We changed presumptive TB cases to “bacteriologically confirmed TB patients”

Line 266-267: There are a number of published studies assessing stool-based TB diagnosis which have used clinical samples (not only spiked samples).

The reviewer is correct. Therefore, we have deleted the word “uniquely” and changed the sentence as follows: Stool samples will not be spiked with mycobacteria as in some other studies (9), (12).

Minor:

Line 70: rather quite- redundancy

We have deleted the word “rather”.

Line 212: Suggest rewording: “Data collected will include age...”

We have changed the sentence accordingly. 

Line 214: “of stool of collection”- delete second “of”

We have deleted the word “of” between “stool” and “collection”.

Reviewer #2: This study protocol is very elaborative and well designed. The authors here would like to emphasize on the need of a simple and standard protocol for Mtb diagnosis from stool samples. Their study protocol is easy to understand with enough supporting information. I would like to provide my concerns/suggestions listed below.

We thank the reviewer for this positive feedback and are pleased to hear that the protocol was easy to understand. We provided the answers to the concerns and suggestions below. 

1. Which sample will be used as baseline? the stool or the sputum or both? 

For the baseline at enrolment, we used both the Xpert-Ultra result of the stool and the sputum specimen depending on the comparison that will be made.

2. Will the samples be collected on two/three consecutive days from the study population or there will be some duration between two samples collected from the same person? My suggestion will be to include this information in the study protocol as well as in the consent form. 

We are pleased that the reviewer is pointing this out as this is indeed an essential detail. We changed the sentence in the method section as follows: Participants will be provided with two (children) or three (adults) large stool containers to allow collection of at least 30 grams of stool on consecutive days. This will indeed be explained to the participants upon providing information about the study. Please note however, that in practice, we will depend on the participant’s willingness to travel back to the facility on consecutive days. For practical reasons, we therefore will allow any sample that is collected within 5 days after starting TB treatment.

3. Are you going to enroll Mtb negative population for this study as a control group? 

We like to thank the reviewer for this suggestion, as we have been doubting taking that approach. However, adding these samples would massively increase the sample size and the cost of the study as MTB won’t be found in the great majority of presumptive TB patients, especially when it concerns children. Besides, the MTB-negative population might only provide limited insights in the main objective of assessing the robustness of the SOS stool method and stool storage and transit conditions. 

Although we agree that there would be benefit in including MTB-negative persons for the sampling strategy experiment, we assume that the chance of finding MTB in one of the aliquots of an (initially) MTB-negative person is very low (<<10%) and we therefore do not include subjects who are negative on both stool and sputum. However, as we will include patients with either sputum/GA or stool MTB positive, we might have some participants for whom the initial stool specimen is MTB negative. 

4. For experiment 2, to study the effect of storage conditions, you have not proposed to freeze any samples. I believe the addition of freezing as a stool storage temperature and then study the effect of freezing on Mtb detection using your proposed SOS method would add value to your studies and also help others in future. 

We agree with the reviewer, and we had originally planned to include storage in freezer as one of the options. However, logistical constraints made us to remove that option. First, our partner in Ethiopia has limited space in their freezers, for which use we also would have to pay. Second, there is limited funding available for this study. Third, the amount of stool collected for the experiments will limit the number of aliquots that we can take from these stools for the experiments. In routine practice, stool storage before processing most likely will occur in the refrigerator or at ambient temperature (which is between 20-25°C in Ethiopia but may be well above 30°C in other countries), rather than in a freezer. That’s why we chose to store stool at (4-6°C, RT, and 36°C). However, to get some insight in the effect of storage of stool in a freezer, we have included in protocol that left-over stool (if any) should be stored in the -20°C freezer.

---

## [Editor Report · Decision Letter 1]

26 Jul 2022

PONE-D-22-03274R1The Simple One-step stool processing method for detection of Pulmonary tuberculosis: a study protocol to assess the robustness, stool storage conditions and sampling strategy for global implementation and scale-upPLOS ONE

Dear Dr. de haas,

Thank you for submitting your manuscript to PLOS ONE. After careful consideration, we feel that it has merit but does not fully meet PLOS ONE’s publication criteria as it currently stands. Therefore, we invite you to submit a revised version of the manuscript that addresses the points raised during the review process.

 The above manuscript is greatly improved int the revised version. I am mostly satisfied with your response to the reviewers concerns. The protocol carries much better clarity. However, right statistics do play a major role in determining the sample size and the analysis of the proposed study. I am not sure if the use of Statulator.com is enough for such studies. I do like to see the study established in consultation with an expert statistician, especially if enrolling a small number of MTB confirmed negative patients might help them establish their specificity. I understand the financial and resource limitations, but strongly recommend it, as this would be very important in understanding the application of the proposed method for routine use. Therefore I am recommending minor revision to the protocol.**********

We look forward to receiving your revised manuscript.

Kind regards,

Padmapriya P Banada, PhD

Academic Editor

PLOS ONE

Journal Requirements:

Additional Editor Comments (if provided):

The above manuscript is greatly improved int the revised version. I am mostly satisfied with your response to the reviewers concerns. The protocol carries much better clarity. However, right statistics do play a major role in determining the sample size and the analysis of the proposed study. I am not sure if the use of Statulator.com is enough for such studies. I do like to see the study established in consultation with an expert statistician, especially if enrolling a small number of MTB confirmed negative patients might help them establish their specificity. I understand the financial and resource limitations, but strongly recommend it, as this would be very important in understanding the application of the proposed method for routine use. Therefore I am recommending minor revision to the protocol.
---

## [Author Response · Author response to Decision Letter 1]

26 Aug 2022

We would like to thank the reviewers for their positive response on the revised version of the manuscript. It was nice to hear that the protocol has greatly improved. We herewith like to provide feedback to the last comment provided by reviewer 1 

“The above manuscript is greatly improved int the revised version. I am mostly satisfied with your response to the reviewer’s concerns. The protocol carries much better clarity. However, right statistics do play a major role in determining the sample size and the analysis of the proposed study. I am not sure if the use of Statulator.com is enough for such studies. I do like to see the study established in consultation with an expert statistician, especially if enrolling a small number of MTB confirmed negative patients might help them establish their specificity. I understand the financial and resource limitations, but strongly recommend it, as this would be very important in understanding the application of the proposed method for routine use. Therefore, I am recommending minor revision to the protocol.”

Reply: Though not explicitly stated on their website, Statulator seems to make use of Mcnemar’s two-sample paired proportions test to estimate sample sizes. We consulted Dr L. Stuck at the university of Amsterdam, a statistician, epidemiologist and data scientist (https://www.aighd.org/people/logan-stuck/), who confirmed that this is the correct test to use for sample size calculations. For reproducibility, we repeated the sample size calculations in Stata using McNemar’s test. This can be done using the “power paired proportions” command. We have adapted the title of Figure 4 accordingly.

---

## [Editor Report · Decision Letter 2]

12 Sep 2022

The Simple One-step stool processing method for detection of Pulmonary tuberculosis: a study protocol to assess the robustness, stool storage conditions and sampling strategy for global implementation and scale-up

PONE-D-22-03274R2

Dear Dr. de haas,

We’re pleased to inform you that your manuscript has been judged scientifically suitable for publication and will be formally accepted for publication once it meets all outstanding technical requirements.

Kind regards,

Padmapriya P Banada, PhD

Academic Editor

PLOS ONE
---

## [Editor Report · Acceptance letter]

26 Sep 2022

PONE-D-22-03274R2 

The Simple One-step stool processing method for detection of Pulmonary tuberculosis: a study protocol to assess the robustness, stool storage conditions and sampling strategy for global implementation and scale-up 

Dear Dr. de haas:

I'm pleased to inform you that your manuscript has been deemed suitable for publication in PLOS ONE. Congratulations! Your manuscript is now with our production department. 

Kind regards, 

on behalf of

Dr. Padmapriya P Banada 

Academic Editor

PLOS ONE